# MITIGATING ADVERSARIAL EFFECTS THROUGH RANDOMIZATION

**Cihang Xie, Zhishuai Zhang & Alan L. Yuille**
Department of Computer Science
The Johns Hopkins University
Baltimore, MD 21218 USA
{cihangxie306, zhshuai.zhang, alan.l.yuille}@gmail.com

**Jianyu Wang**
Baidu Research USA
Sunnyvale, CA 94089 USA
wjyouch@gmail.com

**Zhou Ren**
Snap Inc.
Venice, CA 90291 USA
zhou.ren@snapchat.com

## ABSTRACT

Convolutional neural networks have demonstrated high accuracy on various tasks in recent years. However, they are extremely vulnerable to adversarial examples. For example, imperceptible perturbations added to clean images can cause convolutional neural networks to fail. In this paper, we propose to utilize randomization at inference time to mitigate adversarial effects. Specifically, we use two randomization operations: random resizing, which resizes the input images to a random size, and random padding, which pads zeros around the input images in a random manner. Extensive experiments demonstrate that the proposed randomization method is very effective at defending against both single-step and iterative attacks. Our method provides the following advantages: 1) no additional training or fine-tuning, 2) very few additional computations, 3) compatible with other adversarial defense methods. By combining the proposed randomization method with an adversarially trained model, it achieves a normalized score of 0.924 (ranked No.2 among 107 defense teams) in the NIPS 2017 adversarial examples defense challenge, which is far better than using adversarial training alone with a normalized score of 0.773 (ranked No.56). The code is public available at https://github.com/cihangxie/NIPS2017_adv_challenge_defense.

## 1 INTRODUCTION

Convolutional Neural Networks (CNNs) have been successfully applied to a wide range of vision tasks, including image classification (Krizhevsky et al., 2012; Simonyan & Zisserman, 2015; He et al., 2016a), object detection (Girshick, 2015; Ren et al., 2015; Zhang et al., 2017), semantic segmentation (Long et al., 2015; Chen et al., 2017), visual concept discovery (Wang et al., 2017) etc. However, recent works show that CNNs are extremely vulnerable to small perturbations to the input image. For example, adding visually imperceptible perturbations to the original image can result in failures for image classification (Szegedy et al., 2014; Goodfellow et al., 2015), object detection (Xie et al., 2017) and semantic segmentation (Xie et al., 2017; Fischer et al., 2017; Cisse et al., 2017). These perturbed images are called adversarial examples and Figure 1 gives an example. Adversarial examples pose a great security danger to the deployment of commercial machine learning systems. Thus, making CNNs more robust to adversarial examples is a very important yet challenging problem. Recent works (Papernot et al., 2016b; Kurakin et al., 2017; Tramèr et al., 2017; Cao & Gong,

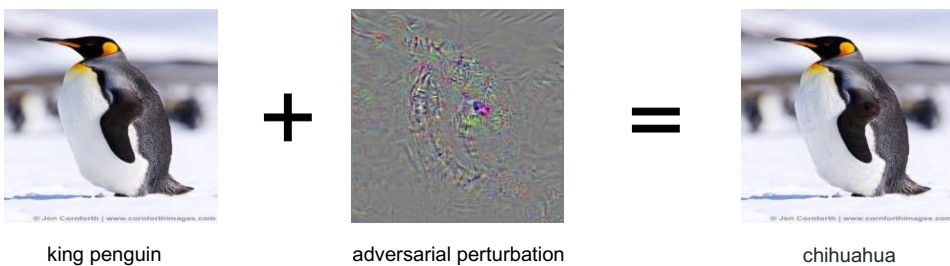

king penguin          adversarial perturbation          chihuahua

Figure 1: This is an adversarial example crafted for VGG (Simonyan & Zisserman, 2015). The left image is classified correctly as king penguin, the center image is the adversarial perturbation (magnified by 10 and enlarged by 128 for better visualization), and the right image is the adversarial example misclassfied as chihuahua.

2017; Metzen et al., 2017; Feinman et al., 2017; Meng & Chen, 2017) are making progress on this line of research.

Adversarial attacks can be divided into two categories: single-step attacks, which perform only one step of gradient computation, and iterative attacks, which perform multiple steps. Intuitively, the perturbation generated by iterative methods may easily get over-fitted to the specific network parameters, and thus be less transferable. On the other hand, single-step methods may not be strong enough to fool the network. For examples, it has been demonstrated that single-step attacks, like Fast Gradient Sign Method (FGSM) (Goodfellow et al., 2015), have better transferability but weaker attack rate than iterative attacks, like DeepFool (Moosavi-Dezfooli et al., 2016).

Due to the weak generalization of iterative attacks, low-level image transformations, e.g., resizing, padding, compression, etc, may probably destroy the specific structure of adversarial perturbations, thus making it a good defense. It can even defend against white-box iterative attacks if random transformations are applied. This is because each test image goes through random transformations and the attacker does not know the specific transformation when generating adversarial noise. Recently, adversarial training (Kurakin et al., 2017; Tramèr et al., 2017) was developed to defend against single-step attacks. Thus by adding the proposed random transformations as additional layers to an adversarially trained model (Tramèr et al., 2017), it is expected that the method is able to effectively defend against both single-step and iterative attacks, including both black-box and white-box settings.

Based on the above reasoning, in this paper, we propose a defense method by randomization at inference time, i.e., random resizing and random padding, to mitigate adversarial effects. To the best of our knowledge, this is the first work that demonstrates the effectiveness of randomization at inference time on mitigating adversarial effects on large-scale dataset, e.g., ImageNet (Deng et al., 2009). The proposed method provides the following advantages:

- Randomization at inference time makes the network much more robust to adversarial images, especially for iterative attacks (both white-box and black box), but hardly hurts the performance on clean (non-adversarial) images. Experiments on section 4.2 support this argument.

- There is no additional training or fine-tuning required which is easy for implementation.

- Very few computations are required by adding the two randomization layers, thus there is nearly no run time increase.

- Randomization layers are compatible to different network structures and adversarial defense methods, which can serve as a basic network module for adversarial defense.

We conduct comprehensive experiments to test the effectiveness of our defense method, using different network structures, against different attack methods, and under different attack scenarios. The results in Section 4 demonstrate that the proposed randomization layers can significantly mitigate adversarial effects, especially for iterative attack methods. Moreover, we submitted the model, which combines the proposed randomization layers and an adversarially trained model (Tramèr et al., 2017), to the NIPS 2017 adversarial examples defense challenge. It reaches a normalized

score of 0.924 (ranked No.2 among 107 defense teams), which is far better than just using adversarial training (Tramèr et al., 2017) alone with a normalized score of 0.773 (ranked No.56).

## 2 RELATED WORK

### 2.1 GENERATING ADVERSARIAL EXAMPLES

Generating adversarial examples has been extensively studied recently. (Szegedy et al., 2014) first showed that adversarial examples, computed by adding visually imperceptible perturbations to the original images, make CNNs predict wrong labels with high confidence. (Goodfellow et al., 2015) proposed the fast gradient sign method to generate adversarial examples based on the linear nature of CNNs, and also proposed adversarial training for defense. (Moosavi-Dezfooli et al., 2016) generated adversarial examples by assuming that the loss function can be linearized around the current data point at each iteration. (Carlini & Wagner, 2017) developed a stronger attack to find adversarial perturbations by introducing auxiliary variables which incooperate the pixel value constrain, e.g., pixel intensity must be within the range [0,255], naturally into the loss function and make the optimization process easier. (Liu et al., 2017) proposed an ensemble-based approaches to generate adversarial examples with stronger transferability. Unlike the works above, (Biggio & Laskov, 2012; Koh & Liang, 2017) showed that manipulating only a small fraction of the training data can significantly increase the number of misclassified samples at test time for learning algorithms, and such attacks are called poisoning attacks.

### 2.2 DEFENDING AGAINST ADVERSARIAL EXAMPLES

Opposite to generating adversarial examples, there is also progress on reducing the effects of adversarial examples. (Papernot et al., 2016b) showed networks trained using defensive distillation can effectively defend against adversarial examples. (Kurakin et al., 2017) proposed to replace the original clean images with a mixture of clean images and corresponding adversarial images in each training batch to improve the network robustness. (Tramèr et al., 2017) improved the robustness further by training the network on an ensemble of adversarial images generated from the trained model itself and from a number of other pre-trained models. Cao & Gong (2017) proposed a region-based classification to let models be robust to adversarial examples. (Metzen et al., 2017) trained a detector on the inner layer of the classifier to detect adversarial examples. (Feinman et al., 2017) detected adversarial examples by looking at the Bayesian uncertainty estimates of the input images in dropout neural networks and by performing density estimation in the subspace of deep features learned by the model. MagNet (Meng & Chen, 2017) detected adversarial examples with large perturbation using detector networks, and pushed adversarial examples with small perturbation towards the manifold of clean images.

## 3 APPROACH

### 3.1 AN OVERVIEW OF GENERATING ADVERSARIAL EXAMPLES

Before introducing the proposed adversarial defense method, we give an overview of generating adversarial examples. Let $X_n$ denote the $n$-th image in a dataset containing $N$ images, and let $y_n^{\text{true}}$ denote the corresponding ground-truth label. We use $\theta$ to denote the network parameters, and $L(X_n, y_n^{\text{true}}; \theta)$ to denote the loss. For the adversarial example generation, the goal is to maximize the loss $L(X_n + r_n, y_n^{\text{true}}; \theta)$ for each image $X_n$, under the constraint that the generated adversarial example $X_n^{\text{adv}} = X_n + r_n$ should look visually similar to the original image $X_n$, i.e., $||r_n|| \leq \epsilon$, and the corresponding predicted label $y_n^{\text{adv}} \neq y_n^{\text{true}}$.

In our experiment, we consider three different attack methods, including one single-step attack method and two iterative attack methods. We use the cleverhans library (Papernot et al., 2016a) to generate adversarial examples, where all these attacks have been implemented via TensorFlow.

- Fast Gradient Sign Method (FGSM): FGSM (Goodfellow et al., 2015) is a single-step attack method. It finds the adversarial perturbation that yields the highest increase of the

linear cost function under $l_\infty$-norm. The update equation is

$$X_n^{\text{adv}} = X_n + \epsilon \cdot sign(\nabla_{X_n} L(X_n, y_n^{\text{true}}; \theta)),  \tag{1}$$

where $\epsilon$ controls the magnitude of adversarial perturbation. In the experiment, we choose $\epsilon = \{2, 5, 10\}$, which corresponds to small, medium and high magnitude of adversarial perturbations, respectively.

- DeepFool: DeepFool (Moosavi-Dezfooli et al., 2016) is an iterative attack method which finds the minimal perturbation to cross the decision boundary based on the linearization of the classifier at each iteration. Any $l_p$-norm can be used with DeepFool, and we choose $l_2$-norm for the study in this paper.

- Carlini & Wagner (C&W): C&W (Carlini & Wagner, 2017) is a stronger iterative attack method proposed recently. It finds the adversarial perturbation $r_n$ by using an auxiliary variable $\omega_n$ as

$$r_n = \frac{1}{2}(tanh(\omega_n + 1)) - X_n.  \tag{2}$$

Then the loss function optimizes the auxiliary variable $\omega_n$

$$\min_{\omega_n} ||\frac{1}{2}(tanh(\omega_n) + 1) - X_n|| + c \cdot f(\frac{1}{2}(tanh(\omega_n) + 1)).  \tag{3}$$

The function $f(\cdot)$ is defined as

$$f(x) = \max(Z(x)_{y^{\text{true}}} - \max\{Z(x)_i : i \neq y^{\text{true}}\}, -k),  \tag{4}$$

where $Z(x)_i$ is the logits output for class $i$, and $k$ controls the confidence gap between the adversarial class and true class. C&W can also work with various $l_p$-norm, and we choose $l_2$-norm in the experiments.

## 3.2 DEFENDING AGAINST ADVERSARIAL EXAMPLES

The goal of defense is to build a network that is robust to adversarial examples, i.e., it can classify adversarial images correctly with little performance loss on non-adversarial (clean) images. Towards this goal, we propose a randomization-based method, as shown in Figure 2, which adds a random resizing layer and a random padding layer to the beginning of the classification networks. There is no re-training or fine-tuning needed which makes the proposed method very easy to implement.

### 3.2.1 RANDOMIZATION LAYERS

The first randomization layer is a random resizing layer, which resizes the original image $X_n$ with the size $W \times H \times 3$ to a new image $X_n'$ with random size $W' \times H' \times 3$. Note that, $|W' - W|$ and $|H' - H|$ should be within a reasonably small range, otherwise the network performance on non-adversarial images would significantly drop. Taking Inception-ResNet network (Szegedy et al., 2017) as an example, the original data input size is $299 \times 299 \times 3$. Empirically we found that the network performance hardly drops if we control the height and width of the resized image $X_n'$ to be within the range $[299, 331)$.

The second randomization layer is the random padding layer, which pads zeros around the resized image in a random manner. Specifically, by padding the resized image $X_n'$ into a new image $X_n''$ with the size $W'' \times H'' \times 3$, we can choose to pad $w$ zero pixels on the left, $W'' - W' - w$ zero pixels on the right, $h$ zero pixels on the top and $H'' - H' - h$ zero pixels on the bottom. This results in a total number of $(W'' - W' + 1) \times (H'' - H' + 1)$ different possible padding patterns.

During implementation, the original image first goes through two randomization layers, and then we pass the transformed image to the original CNN for classification. The pipeline is illustrated in Figure 2.

### 3.2.2 RANDOMIZATION LAYERS + ADVERSARIAL TRAINING

Note that our randomization-based method is good at defending against iterative attacks, and adversarial training (Kurakin et al., 2017; Tramèr et al., 2017) can effectively increase the robustness of neural networks to single-step attacks. Thus, to make the best of both worlds, we can combine the proposed randomization layers and an adversarially trained model (Tramèr et al., 2017) together to defend against both single-step and iterative attacks.

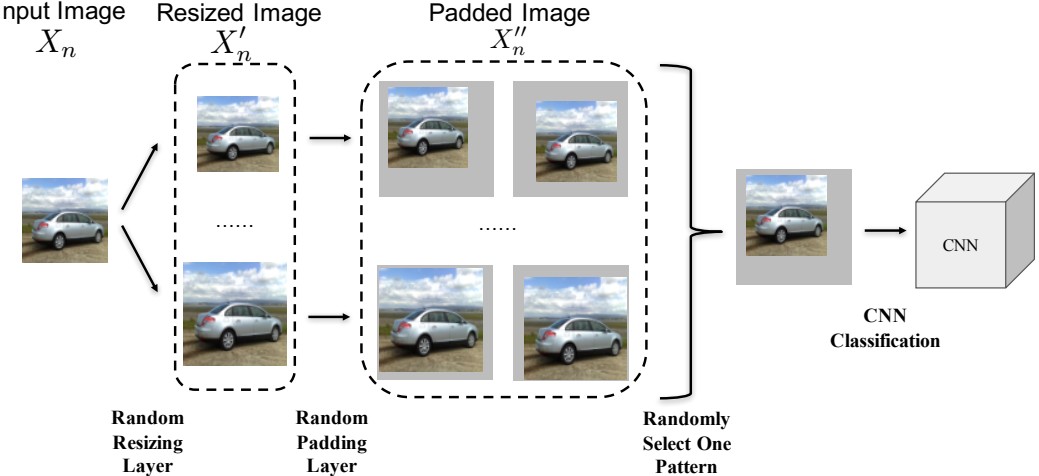

Figure 2: The pipeline of our randomization-based defense mechanism. The input image $X_n$ first goes through the random resizing layer with **a** random scale applied. Then the random padding layer pads the resized image $X'_n$ in **a** random manner. The resulting padded image $X''_n$ is used for classification.

## 4 EXPERIMENTS

### 4.1 EXPERIMENT SETUP

**Dataset:** It is less meaningful to attack the images that are already classified wrongly. Therefore, we randomly choose 5000 images from the ImageNet validation set that are classified correctly by all the considered networks to form our test dataset. All these images are of the size $299 \times 299 \times 3$.

**Networks:** We test with four publicly available networks[1],[2], including *Inception-v3* (Szegedy et al., 2016), *ResNet-v2* (He et al., 2016b) of 101 layers, *Inception-ResNet-v2* (Szegedy et al., 2017), and *ens-adv-Inception-ResNet-v2* which applies the ensemble adversarial training (Tramèr et al., 2017) on *Inception-ResNet-v2*. These networks have been trained on ImageNet, and **we do not perform any re-training or fine-tuning on them for the whole experiments**.

**Defense Models:** The defense models consist of the original networks (i.e., four above-mentioned networks) and two additional randomization layers. For the random resizing layer, it changes the input shape from $299 \times 299 \times 3$ to $rnd \times rnd \times 3$, where $rnd$ is a integer randomly sampled from the range $[299, 331)$. For the random padding layer, it pads the resized image to the shape of $331 \times 331 \times 3$ in a random manner. By applying these two randomization layers, we can create $\sum_{rnd=299}^{330} (331 - rnd + 1)^2 = 12528$ different patterns for a single image. Since there exists small variance on model performance w.r.t. different random patterns, we run the defense model three times independently and report the average accuracy.

**Target Models under Different Attack Scenarios:** The strongest attack would be that the attackers consider **ALL** possible patterns of the defense models when generating the adversarial examples. However, this is computationally impossible, because failing a large number of patterns (e.g., 12528 here) at the same time takes extremely long time, and may not even converge. Thus, we let attackers use the target models to generate adversarial examples instead, and consider the following three different attack scenarios.

- **Vanilla Attack**: The attackers do not know the existence of the randomization layers and the **target model** is just the original network.

---

[1]https://github.com/tensorflow/models/tree/master/research/slim
[2]http://download.tensorflow.org/models/ens_adv_inception_resnet_v2_2017_08_18.tar.gz

Table 1: Top-1 classification accuracy on the clean images. We see that adding random resizing and random padding cause very little accuracy drop on clean (non-adversarial) images.

| Models | Inception-v3 | ResNet-v2-101 | Inception-ResNet-v2 | ens-adv-Inception-ResNet-v2 |
|---|---|---|---|---|
| w/o randomization layers | 100% | 100% | 100% | 100% |
| w randomization layers | 97.3% | 98.3% | 99.3% | 99.2% |

- **Single-Pattern Attack**: The attackers know the existence of the randomization layers. In order to mimic the structures of defense models, the **target model** is chosen as the original network + randomization layers with only one predefined pattern.

- **Ensemble-Pattern Attack**: The attackers know the existence of the randomization layers. In order to mimic the structures of defense models in a more representative way, the **target model** is chosen as the original network + randomization layers with an ensemble of predefined patterns.

**Target Models and Defense Models:** The target models and the defense models are exactly the same except for the parameter settings of the randomization layers, i.e., the randomization parameters at the target models are predefined while randomization parameters at the defense models are randomly generated at test time. The original networks (e.g., *Inception-v3*) utilized by the target models and the defense models are the same. The attackers first generate adversarial examples using the target models, and then evaluate the classification accuracy of these generated adversarial examples on both the target and defense models. A low accuracy of the target model indicates that the attack is successful, and a high accuracy of the defense model indicates that the defense is effective.

## 4.2    CLEAN IMAGES

Table 1 shows the top-1 accuracy of networks with and without randomization layers on the clean images. We can see that randomization layers introduce negligible performance degradation on clean images. Specifically, we can observe that: (1) models with more advanced architectures tend to have less performance degradation, e.g., *Inception-ResNet-v2* only has $0.7\%$ degradation while *Inception-v3* has $2.7\%$ degradation; (2) ensemble adversarial training brings nearly no performance degradation to the models, e.g., *Inception-ResNet-v2* and *ens-adv-Inception-ResNet-v2* have nearly the same performance degradation.

## 4.3    VANILLA ATTACK SCENARIO

For the vanilla attack scenario, the attackers are not aware of randomization layers, and directly use the original networks as the target model to generate adversarial examples. The attack ability on the defense models mostly rely on the transferability of adversarial examples to different resizing and padding. From the top-1 accuracy presented in Table 2, we observe that randomization layers can mitigate the adversarial effects for both single-step and iterative attacks significantly. As for single-step attacks FGSM-$\epsilon$, larger $\epsilon$ indicates stronger transferability, thus making it harder to defend. However, we can still get satisfactory accuracy of the defense model on single-step attacks (even with large $\epsilon$) using *ens-adv-Inception-ResNet-v2* ($94.3\%$ top-1 accuracy). As for iterative attacks, attackers always reach a very high attack rate on target model, but have almost no impact on models after randomization layers are applied. This is because iterative attack methods are over-fitted to the target models thus have weak transferability.

## 4.4    SINGLE-PATTERN ATTACK SCENARIO

For the single-pattern attack scenario, the attackers are aware of the existence of randomization layers and also the parameters of the random resizing and random padding (i.e., from $299 \times 299$ to $331 \times 331$), but they do not know the specific randomization patterns utilized by the defense models (even the defense models themselves do not know these specific randomization patterns since they are randomly instantiated at test time). In order to generate adversarial examples, the

Table 2: Top-1 classification accuracy under the vanilla attack scenario. We see that randomization layers effectively mitigate adversarial effects for all attacks and all networks. Particularly, combining randomization layers with ensemble adversarial training (*ens-adv-Inception-ResNet-v2*) performs very well on all attacks.

| Models | Inception-v3 | | ResNet-v2-101 | | Inception-ResNet-v2 | | ens-adv-Inception-ResNet-v2 | |
|---|---|---|---|---|---|---|---|---|
| | target model | defense model | target model | defense model | target model | defense model | target model | defense model |
| FGSM-2 | 33.2% | 65.1% | 26.3% | 71.8% | 65.3% | 81.0% | 84.4% | 95.7% |
| FGSM-5 | 31.1% | 54.5% | 20.4% | 54.3% | 61.7% | 74.1% | 87.4% | 94.5% |
| FGSM-10 | 33.0% | 52.4% | 20.4% | 46.1% | 61.2% | 71.3% | 90.2% | 94.3% |
| DeepFool | 0% | 98.3% | 0% | 97.7% | 0% | 98.2% | 0.2% | 99.1% |
| C&W | 0% | 96.9% | 0% | 97.1% | 0.3% | 97.7% | 0.9% | 98.8% |

Table 3: Top-1 classification accuracy under the single-pattern attack scenario. We see that randomization layers effectively mitigate adversarial effects for all attacks and all networks. Particularly, combining randomization layers with ensemble adversarial training (*ens-adv-Inception-ResNet-v2*) performs very well on all attacks.

| Models | Inception-v3 | | ResNet-v2-101 | | Inception-ResNet-v2 | | ens-adv-Inception-ResNet-v2 | |
|---|---|---|---|---|---|---|---|---|
| | target model | defense model | target model | defense model | target model | defense model | target model | defense model |
| FGSM-2 | 35.1% | 63.8% | 29.5% | 70.1% | 71.6% | 83.4% | 86.3% | 96.4% |
| FGSM-5 | 32.4% | 53.9% | 23.2% | 52.3% | 68.3% | 78.2% | 88.4% | 95.4% |
| FGSM-10 | 34.7% | 51.8% | 22.4% | 43.8% | 66.8% | 75.6% | 90.7% | 95.2% |
| DeepFool | 1.1% | 98.2% | 1.7% | 97.8% | 0.6% | 98.4% | 1.0% | 99.2% |
| C&W | 1.1% | 97.4% | 1.7% | 97.0% | 0.8% | 97.9% | 1.6% | 99.1% |

attackers choose the target models as the original networks + randomization layers but with only **one** specific pattern to compute the gradient. In this experiment, the specific pattern that we use is to place the original input $X_n$ at the center of the padded image $X_n''$, i.e., no resizing is applied, and 16 zeros pixels are padded on the left, right, top and bottom on the input images, respectively. Table 3 shows the top-1 accuracy of both target models and defense models, and similar results to vanilla attack scenario are observed: (1) for single-step attacks, randomization layers are less effective on mitigating adversarial effects for a larger $\epsilon$, while the adversarially trained models are able to defend against such attacks; (2) for iterative attacks, they reach high attack rates on target models, while have nearly no impact on defense models.

## 4.5 ENSEMBLE-PATTERN ATTACK SCENARIO

For the ensemble-pattern attack scenario, similar to single-pattern attack scenario, the attackers are aware of the randomization layers and the parameters of the random resizing and random padding (i.e., starting from $299 \times 299$ to $331 \times 331$), but they do not know the specific patterns utilized by the defense models at test time. The target models thus are constructed in a more representative way: let randomization layers choose an ensemble of predefined patterns, and the goal of the attackers is to let all chosen patterns fail on classification. In this experiment, the specific ensemble patterns that we choose are: (1) first resize the input image to five different scales $\{299, 307, 315, 323, 331\}$; (2) then pad each resized image to five different patterns, where the resized image is placed at the top left, top right, bottom left, bottom right, and center of the padded image, respectively. Since there is only one padding pattern for the resized image with size 331, we can obtain $4 * 5 + 1 = 21$ patterns in total. Due to the large computation amounts introduced by the ensemble-pattern attack scenario, we randomly choose 500 images out of the entire test dataset for this experiment. The top-1 accuracy for the target model here is calculated by summing up the number of correctly classified patterns of each

Table 4: Top-1 classification accuracy under the ensemble-pattern attack scenario. Similar to vanilla attack and single-pattern attack scenarios, we see that randomization layers increase the accuracy under all attacks and networks. This clearly demonstrates the effectiveness of the proposed randomization method on defending against adversarial examples, even under this very strong attack scenario.

| Models | Inception-v3 | | ResNet-v2-101 | | Inception-ResNet-v2 | | ens-adv-Inception-ResNet-v2 | |
|---|---|---|---|---|---|---|---|---|
| | target model | defense model | target model | defense model | target model | defense model | target model | defense model |
| FGSM-2 | 37.3% | 41.2% | 39.2% | 44.9% | 71.5% | 74.3% | 86.2% | 88.9% |
| FGSM-5 | 31.7% | 34.0% | 24.6% | 29.7% | 65.2% | 67.3% | 85.8% | 87.5% |
| FGSM-10 | 30.4% | 32.8% | 18.6% | 21.7% | 62.9% | 64.5% | 86.6% | 87.9% |
| DeepFool | 0.6% | 81.3% | 0.9% | 80.5% | 0.9% | 69.4% | 1.6% | 93.5% |
| C&W | 0.6% | 62.9% | 1.0% | 74.3% | 1.6% | 68.3% | 5.8% | 86.1% |

image over the entire pattern number of all images. For the results presented in Table 4, we can see that the adversarial examples generated under ensemble-pattern attack scenario are much stronger. For single-step attacks, the generated adversarial examples can let the performance of the defense model with an adversarially trained network drop around $8\%$ compared to the performance under vanilla attack and single-pattern attack scenarios, and drop much more on other defense models. For iterative attacks, we observe that the adversarial examples generated by C&W are stronger than those generated by DeepFool, e.g., the defense model with *Inception-v3* has an accuracy of $81.3\%$ on DeepFool, while only has an accuracy of $62.9\%$ on C&W. We argue that this is due to the more advanced loss function (i.e., introduction of auxiliary variable for pixel value control) utilized by C&W than DeepFool. Additionally, the accuracy of defense model on C&W can be improved by utilizing more advanced architecture (e.g., *ResNet-v2-101* has higher accuracy than *Inception-v3*) and applying ensemble adversarial training (e.g., *ens-adv-Inception-ResNet-v2* has higher accuracy than *Inception-ResNet-v2*). For the best defense model that we have, *ens-adv-Inception-ResNet-v2* + randomization layers reaches the top-1 accuracy of $93.5\%$ on DeepFool and $86.1\%$ on C&W, respectively.

## 4.6 Diagnostic Experiment

Due to the large amount of possible patterns introduced by randomization layers, it is hard to analyze the effectiveness of random resizing and random padding precisely. In this section, we limit the freedom of randomization to be a small number (i.e., 4 in random padding and 1 in random resizing) and analyze the effectiveness of these two operations separately. The same $500$ images in section 4.5 are used in this experiment. In addition, the input images for target models and defense models are resized to the shape $330 \times 330 \times 3$ beforehand.

### 4.6.1 One Pixel Padding

For the random padding, there are only $4$ patterns when padding the input images from $330 \times 330 \times 3$ to $331 \times 331 \times 3$. In order to construct a stronger attack, we follow the experiment setup in section 4.5 where 3 chosen patterns are ensembled. Specifically, the target model takes an ensemble of patterns where the original images are at the top left, top right and bottom left (3 patterns) of the padded images, and the defense model takes the last pattern where the original images are at the bottom right of the padded images. Note that, since there is no randomization in the defense model, we only run the defense model once. Table 5 summaries the results, and we can see that: (1) adversarial examples generated by single-step attacks have strong transferability, but still cannot attack the defense model with an adversarially trained model successfully (i.e., the defense model with *ens-adv-Inception-ResNet-v2*); (2) adversarial examples generated by iterative attacks are much less transferable between different padding patterns even when only $4$ different patterns exist. The results demonstrate that creating different padding patterns can effectively mitigate adversarial effects.

Table 5: Top-1 classification accuracy under one pixel padding scenario. This table shows that creating different padding patterns (even 1-pixel padding) can effectively mitigate adversarial effects.

| Models | Inception-v3 | | ResNet-v2-101 | | Inception-ResNet-v2 | | ens-adv-Inception-ResNet-v2 | |
|---|---|---|---|---|---|---|---|---|
| | target model | defense model | target model | defense model | target model | defense model | target model | defense model |
| FGSM-2 | 36.4% | 39.6% | 29.8% | 34.4% | 71.3% | 74.0% | 88.2% | 94.8% |
| FGSM-5 | 33.5% | 36.2% | 22.2% | 26.2% | 68.4% | 71.0% | 92.1% | 94.4% |
| FGSM-10 | 34.5% | 38.8% | 21.3% | 23.6% | 67.4% | 70.4% | 93.7% | 94.0% |
| DeepFool | 0.9% | 97.2% | 0.9% | 95.2% | 0.9% | 87.6% | 1.5% | 99.2% |
| C&W | 0.8% | 70.2% | 0.9% | 76.8% | 1.0% | 79.4% | 2.4% | 98.2% |

Table 6: Top-1 classification accuracy under one pixel resizing scenario. This table shows that resizing image to a different scale (even 1-pixel scale) can effectively mitigate adversarial effects.

| Models | Inception-v3 | | ResNet-v2-101 | | Inception-ResNet-v2 | | ens-adv-Inception-ResNet-v2 | |
|---|---|---|---|---|---|---|---|---|
| | target model | defense model | target model | defense model | target model | defense model | target model | defense model |
| FGSM-2 | 30.8% | 56.2% | 31.6% | 44.6% | 66.2% | 75.0% | 87.6% | 97.2% |
| FGSM-5 | 31.2% | 48.8% | 25.6% | 35.8% | 61.4% | 70.2% | 91.2% | 96.6% |
| FGSM-10 | 36.4% | 51.0% | 23.8% | 32.6% | 62.8% | 68.2% | 94.8% | 95.2% |
| DeepFool | 2.6% | 99.4% | 1.0% | 98.6% | 1.2% | 97.4% | 1.2% | 99.4% |
| C&W | 2.6% | 97.8% | 1.0% | 94.8% | 2.0% | 94.8% | 1.8% | 99.6% |

### 4.6.2 ONE-PIXEL RESIZING

For the random resizing, there is only 1 pattern that exists when the input images are resized from $330 \times 330 \times 3$ to $331 \times 331 \times 3$. The results in Table 6 indicate that resizing the images by only 1 pixel can effectively destroy the transferability of adversarial examples by both single-step and iterative attacks.

## 5 NIPS 2017 ADVERSARIAL EXAMPLES DEFENSE CHALLENGE

We submitted our model to the NIPS 2017 adversarial examples defense challenge[3] for a more comprehensive performance evaluation. The test dataset contains 5000 images which are all of the size $299 \times 299 \times 3$, and their corresponding labels are the same as the ImageNet 1000-class labels. Each defense method are run on all 5000 adversarial images generated against all adversarial attacks. For each correctly classified image, the defense method gets one point. The normalized score for each defense method is computed using the following formula:

$$\text{score} = \frac{1}{M} \sum_{\text{attack} \in A} \sum_{n=1}^{5000} \left[ \text{defense}(\text{attack}(X_n)) = y_n^{\text{true}} \right], \tag{5}$$

where $A$ is the set of all attacks, $M$ is the total number of generated adversarial examples by all attacks, and the function $[\cdot]$ is the indicator function which equals to 1 when the prediction is true.

### 5.1 CHALLENGE RESULTS

The best defense model in our experiments, i.e., randomization layers + *ens-adv-Inception-Resnet-v2*, was submitted to the challenge. To increase the classification accuracy, we (1) changed the

---

[3]https://www.kaggle.com/c/nips-2017-defense-against-adversarial-attack

resizing range from $[299, 331)$ to $[310, 331)$; (2) averaged the prediction results over 30 randomization patterns for each image; (3) flipped the input image with probability $0.5$ for each randomization pattern.

By evaluating our model against 156 different attacks, it reaches a normalized score of $0.924$ (ranked No.2 among 107 defense models), which is far better than using ensemble adversarial training (Tramèr et al., 2017) alone with a normalized score of $0.773$ (ranked No.56). This result further demonstrates that the proposed randomization method effectively make deep networks much more robust to adversarial attacks.

## 6 CONCLUSION

In this paper, we propose a randomization-based mechanism to mitigate adversarial effects. We conduct comprehensive experiments to validate the effectiveness of our defense method, using different network structures, against different attack methods, and under different attack scenarios. The experimental results show that adversarial examples rarely transfer between different randomization patterns, especially for iterative attacks. In addition, the proposed randomization layers are compatible to different network structures and adversarial defense methods, which can serve as a basic module for defense against adversarial examples. By adding the proposed randomization layers to an adversarially trained model (Tramèr et al., 2017), it achieves a normalized score of $0.924$ (ranked No.2 among 107 defense models) in the NIPS 2017 adversarial examples defense challenge, which is far better than using adversarial training alone with a normalized score of $0.773$ (ranked No.56). The code is public available at `https://github.com/cihangxie/NIPS2017_adv_challenge_defense`.

## ACKNOWLEDGEMENTS

This work is supported by a gift grant from SNAP Research, ONR–N00014-15-1-2356 and NSF Visual Cortex on Silicon CCF-1317560.

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

## APPENDIX A    OTHER RANDOMIZATION METHODS

Besides random resizing and random padding, we further evaluate the effectiveness of four other randomization methods against adversarial examples. All these four methods are used as data-augmentation during the standard network training.

- Random Brightness: a brightness factor $\delta$ is randomly picked in the interval $[-\delta_{\max}, \delta_{\max}]$ to adjust the brightness of the normalized image $\hat{X}_n$. We choose $\delta_{\max} = \frac{32}{255}$.
- Random Saturation:   a saturation factor $\alpha$ is randomly picked in the interval $[\alpha_{lower}, \alpha_{upper}]$ to adjust the saturation of the normalized image $\hat{X}_n$. We choose $\alpha_{lower} = 0.5$ and $\alpha_{upper} = 1.5$.
- Random Hue: a hue factor $\theta$ is randomly picked in the interval $[-\theta_{\max}, \theta_{\max}]$ to adjust the hue of the normalized image $\hat{X}_n$. We choose $\theta_{\max} = 0.2$.
- Random Contrast: a contrast factor $\beta$ is randomly picked in the interval $[\beta_{lower}, \beta_{upper}]$ to adjust the contrast of the normalized image $\hat{X}_n$. We choose $\beta_{lower} = 0.5$ and $\beta_{upper} = 1.5$.

Note that, (1) the parameters chosen above are the same as the ones used during network training process[4]; (2) the pixel value of the normalized image $\hat{X}_n$ are all within the interval $[0, 1]$, and we also use this range to clip the pixel value of the image after pre-processing.

Following the experiment setup in section 4, we first evaluate the effectiveness of each of these randomization methods on the 5000 clean images. The results are shown in the Table 7. We can see that these methods hardly hurt the performance on clean images. We further combine the proposed randomization layers, i.e., random resizing layer and random padding layer, with each of these randomization methods (denoted as "++"). We see that the combined randomization modules only cause very little accuracy drop on clean images.

Table 7: Top-1 classification accuracy on clean images. We see that these four randomization methods hardly hurt the performance on clean images. **We use "++" to denote the addition of the proposed randomization layers**, i.e., random resizing and random padding, and the results indicate that combined models still performs pretty good on clean images.

| Models | Inception-v3 | ResNet-v2-101 | Inception-ResNet-v2 | ens-adv-Inception-ResNet-v2 |
|---|---|---|---|---|
| random brightness | 99.6% | 99.7% | 99.8% | 99.8% |
| random brightness ++ | 98.6% | 98.1% | 99.1% | 99.2% |
| random saturation | 99.6% | 99.7% | 99.9% | 99.9% |
| random saturation ++ | 98.6% | 98.3% | 99.3% | 99.3% |
| random hue | 99.4% | 99.6% | 99.7% | 99.4% |
| random hue ++ | 98.6% | 98.3% | 99.2% | 99.1% |
| random contrast | 99.5% | 99.6% | 99.7% | 99.6% |
| random contrast ++ | 98.6% | 98.2% | 99.3% | 99.1% |

We then evaluate the effectiveness of these randomization methods against the adversarial examples generated under the vanilla attack scenario. The results are shown in the Tables 8 - 11. Compared to the results in Table 2, all these four methods are much less effective than the proposed randomization layers. By combining the proposed randomization layers with each of these four randomization methods (denoted as "++"), the performance can be slightly improved than using the proposed randomization layers alone.

Since each of these four randomization methods alone are not as effective as our proposed randomization methods against adversarial examples generated under the vanilla attack scenario, we do not further investigate their effectiveness under single-pattern attack and ensemble-pattern attack scenarios. However, combining these randomization methods with our proposed randomization layers together provides a way to build a slightly stronger defense mechanism.

---

[4]https://github.com/tensorflow/models/blob/master/research/inception/inception/image_processing.py#L182

Table 8: Top-1 classification accuracy by using random brightness under the vanilla attack scenario. Compared to the results in Table 2, random brightness is much less effective than the proposed randomization layers. By combing random brightness and the proposed randomization layers (denoted as **random brightness++**), it reaches slightly better performance than using the proposed randomization layers alone.

| Models | Inception-v3 | | ResNet-v2-101 | | Inception-ResNet-v2 | | ens-adv-Inception-ResNet-v2 | |
|---|---|---|---|---|---|---|---|---|
| | random brightness | random brightness++ | random brightness | random brightness++ | random brightness | random brightness++ | random brightness | random brightness++ |
| FGSM-2 | 34.9% | 67.0% | 28.5% | 73.5% | 66.4% | 81.3% | 85.0% | 95.8% |
| FGSM-5 | 31.9% | 55.5% | 21.6% | 55.7% | 62.4% | 74.7% | 87.6% | 95.0% |
| FGSM-10 | 33.2% | 52.9% | 20.9% | 47.2% | 61.8% | 71.5% | 90.4% | 94.5% |
| DeepFool | 79.4% | 98.1% | 82.3% | 97.5% | 62.8% | 98.3% | 79.0% | 99.1% |
| C&W | 34.5% | 96.9% | 47.7% | 97.2% | 51.3% | 98.0% | 42.3% | 98.6% |

Table 9: Top-1 classification accuracy by using random saturation under the vanilla attack scenario. Compared to the results in Table 2, random saturation is much less effective than the proposed randomization layers. By combing random saturation and the proposed randomization layers (denoted as **random saturation++**), it reaches slightly better performance than using the proposed randomization layers alone.

| Models | Inception-v3 | | ResNet-v2-101 | | Inception-ResNet-v2 | | ens-adv-Inception-ResNet-v2 | |
|---|---|---|---|---|---|---|---|---|
| | random saturation | random saturation++ | random saturation | random saturation++ | random saturation | random saturation++ | random saturation | random saturation++ |
| FGSM-2 | 34.2% | 66.5% | 27.9% | 73.6% | 66.3% | 81.5% | 85.2% | 95.7% |
| FGSM-5 | 31.8% | 55.2% | 21.1% | 55.7% | 62.1% | 74.6% | 87.0% | 95.0% |
| FGSM-10 | 33.5% | 52.2% | 20.7% | 46.5% | 61.7% | 71.4% | 90.1% | 93.9% |
| DeepFool | 82.6% | 98.1% | 79.6% | 97.6% | 64.7% | 98.2% | 78.5% | 99.0% |
| C&W | 39.2% | 97.2% | 47.5% | 96.9% | 51.7% | 97.7% | 50.9% | 99.1% |

Table 10: Top-1 classification accuracy by using random hue under the vanilla attack scenario. Compared to the results in Table 2, random hue is much less effective than the proposed randomization layers. By combing random hue and the proposed randomization layers (denoted as **random hue++**), it reaches slightly better performance than using the proposed randomization layers alone.

| Models | Inception-v3 | | ResNet-v2-101 | | Inception-ResNet-v2 | | ens-adv-Inception-ResNet-v2 | |
|---|---|---|---|---|---|---|---|---|
| | random hue | random hue++ | random hue | random hue++ | random hue | random hue++ | random hue | random hue++ |
| FGSM-2 | 38.1% | 69.0% | 32.0% | 74.9% | 68.6% | 83.0% | 87.4% | 95.8% |
| FGSM-5 | 33.9% | 57.2% | 23.0% | 57.6% | 64.0% | 76.1% | 86.7% | 93.9% |
| FGSM-10 | 36.4% | 54.2% | 22.1% | 48.4% | 63.0% | 72.5% | 88.0% | 91.3% |
| DeepFool | 95.0% | 97.9% | 91.2% | 97.6% | 86.5% | 98.4% | 96.8% | 99.1% |
| C&W | 72.1% | 97.3% | 74.0% | 97.0% | 77.4% | 98.2% | 81.1% | 98.8% |

Table 11: Top-1 classification accuracy by using random contrast under the vanilla attack scenario. Compared to the results in Table 2, random contrast is much less effective than the proposed randomization layers. By combing random contrast and the proposed randomization layers (denoted as **random contrast++**), it reaches slightly better performance than using the proposed randomization layers alone.

| Models | Inception-v3 | | ResNet-v2-101 | | Inception-ResNet-v2 | | ens-adv-Inception-ResNet-v2 | |
|---|---|---|---|---|---|---|---|---|
| | random contrast | random contrast++ | random contrast | random contrast++ | random contrast | random contrast++ | random contrast | random contrast++ |
| FGSM-2 | 37.0% | 68.0% | 29.4% | 74.1% | 67.2% | 82.5% | 85.9% | 96.0% |
| FGSM-5 | 32.7% | 56.3% | 22.7% | 57.0% | 63.1% | 74.9% | 88.1% | 94.9% |
| FGSM-10 | 34.4% | 53.5% | 21.2% | 47.0% | 62.1% | 72.0% | 90.6% | 94.3% |
| DeepFool | 90.4% | 98.1% | 87.5% | 97.5% | 73.8% | 98.1% | 90.8% | 99.0% |
| C&W | 56.8% | 97.0% | 57.1% | 96.7% | 63.7% | 97.9% | 67.6% | 98.8% |

## APPENDIX B RANDOMIZATION LAYERS WITH SMALLER SIZE

Instead of resizing the input image to a larger size, we here resize the input image to a smaller size, i.e., the resizing parameter is randomly sampled from the range $[267, 299)$. The random padding layer then pads the resized image to the shape of $299 \times 299 \times 3$ in a random manner. Note that, the random resizing layer and the random padding layer here have the same freedom as the ones used in the paper, i.e., they create the same number, 12528, of different patterns for a single image. We evaluate the effectiveness of this parameter setting on both the 5000 clean images and the adversarial examples generated under the vanilla attack scenario. The results are shown in the Table 12. We see that randomization layers still work well with smaller size images, but is slightly worse than using larger size images as in the paper. This is because resizing to a smaller size loses certain information of the original image.

Table 12: Top-1 classification accuracy on the clean images and the adversarial examples generated under the vanilla attack scenario. Compared to the results in Tables 1 and 2, randomization parameters applied here (i.e., resize between $[267, 299)$, and pad to $299 \times 299 \times 3$) is slightly worse than the randomization parameters applied in the paper (i.e., resize between $[299, 331)$, and pad to $331 \times 331 \times 3$).

| Models | Inception-v3 | ResNet-v2-101 | Inception-ResNet-v2 | ens-adv-Inception-ResNet-v2 |
|---|---|---|---|---|
| clean images | 98.2% | 97.5% | 99.1% | 98.7% |
| FGSM-2 | 63.1% | 65.0% | 79.9% | 95.0% |
| FGSM-5 | 53.4% | 48.3% | 73.3% | 94.0% |
| FGSM-10 | 50.8% | 40.5% | 70.6% | 93.4% |
| DeepFool | 97.2% | 96.5% | 96.0% | 98.6% |
| C&W | 95.2% | 95.2% | 97.2% | 97.8% |

## APPENDIX C RANDOMIZATION LAYERS WITH MULTIPLE ITERATIONS

In this section, we show the relationship between the top-1 accuracy of the defense model and the iteration number performed on each image. Specifically, we choose *ens-adv-Inception-ResNet-v2 +* randomization layers as the defense model for the experiment. The same trend can be observed for other defense models.

For the defense model, the iteration number is chosen to be $\{1, 5, 10, 20, 30\}$, and it is evaluated on the 5000 clean test images and the adversarial examples generated under all three attack scenarios. The results are shown in the Figures 3 - 5. We can observe that (1) increasing the number of

iteration can slightly improve the top-1 classification accuracy of the defense model on clean images and adversarial examples generated under both the vanilla attack and the single-pattern attack scenarios; (2) increasing the number of iteration has nearly no improvement for the top-1 classification accuracy of the defense model on adversarial examples generated by single-step attacks under the ensemble-pattern attack scenario; (3) increasing the number of iteration can improve the top-1 classification accuracy of the defense model on adversarial examples generated by iterative attacks under the ensemble-pattern attack scenario.

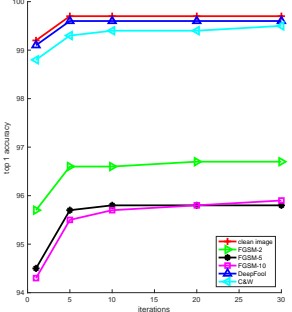 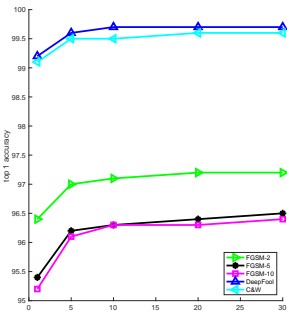 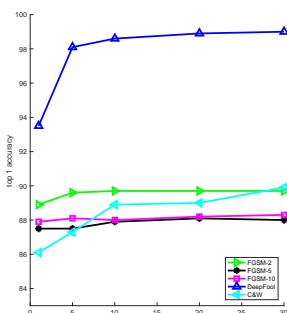

Figure 3: Top-1 classification accuracy on the clean images and the adversarial examples generated under the vanilla attack scenrio.

Figure 4: Top-1 classification accuracy on the adversarial examples generated under the single-pattern attack scenrio.

Figure 5: Top-1 classification accuracy on the adversarial examples generated under the ensemble-pattern attack scenrio.

