# OpenReview forum: "Mitigating Adversarial Effects Through Randomization"
_ICLR.cc/2018/Conference — Accept (Poster)_

### Official Review · AnonReviewer1 · 2017-11-27
**unclear effects of randomization**

**Rating:** 6
**Confidence:** 4

**Review:**

The authors propose a simple defense against adversarial attacks, which is to add randomization in the input of the CNNs. They experiment with different CNNs and published adversarial training techniques and show that randomized inputs mitigate adversarial attacks.

Pros:
(+) The idea introduced is simple and flexible to be used for any CNN architecture
(+) Experiments on ImageNet1k prove demonstrate its effectiveness
Cons:
(-) Experiments are not thorougly explained
(-) Novelty is extremely limited
(-) Some baselines missing


The experimental section of the paper was rather confusing. The authors should explain the experiments and the settings in the table, as those are not very clear. In particular, it was not clear whether the defense model was trained with the input randomization layers? Also, in Tables 1-6, how was the target model trained? How do the training procedures of target vs. defense model differ? In those tables, what is the testing procedure for the target model and how does it compare to the defense model?

The gap between the target and defense model in Table 4 (ensemble pattern attack scenario) shrinks for single step attack methods. This means that when the attacker is aware of the randomization parameters, the effect of randomization might diminish. A baseline that reports the performance when the attacker is fully aware of the randomization of the defender (parameters, patterns etc.) is missing but is very useful.

While the experiments show that the randomization layers mitigate the effect of randomization attacks, it's not clear whether the effectiveness of this very simple approach is heavily biased towards the published ways of generating adversarial attacks and the particular problem (i.e. classification). The form of attacks studied in the paper is that of additive noise. But there is many types of attacks that could be closely related to the randomization procedure of the input and that could lead to very different results.

---

> ### Author Response · Authors · 2017-12-30
> **Thanks for your comments and a clearer version updated**
>
> Thank you very much for the comments. We have updated our paper, especially the experiment section. Below are the detailed answers to your concerns.
>
> “experiment confusing”: Sorry for the confusion, and we have made this clearer in the updated paper. The defense model is simply adding two randomization layers to the beginning of the original classification networks. There is no re-training and fine-tuning needed. This is an advantage of our method. We choose Inception-v3, ResNet-v2, Inception-ResNet-v2 and ens-adv-Inception-ResNet-v2 as the original CNN models, and these models are public available under Tensorflow github repo. The target models are the models used by attackers to generate adversarial examples. The target models differ under different attack scenarios: (1) vanilla attack: the target model is the original CNN model, e.g. Inception-v3; (2) single-pattern attack: target model is the original CNN model + randomization layers with only one predefined pattern; (3) ensemble-pattern attack: the target model is the original CNN model + randomization layers with an ensemble of predefined patterns. Note that the structure and weights of the classification network in target model and defense model are exactly the same. In tables 2-6, the attackers first use target model to generate adversarial examples, and then tests the top-1 accuracy on target model and defense model. Specifically, (1) for target model, a lower accuracy indicates a more successful attack; (2) for defense model, a higher accuracy indicates a more successful defense.
>
> “stronger baseline when the attacker is fully aware the patterns”: We agree that the performance gap between the target and defense model will shrink as more randomization patterns are considered in the attack process. This is expected. Here we want to emphasize that during defense, the padding and resizing are done randomly, so there is no way for both the attacker and the defender to know the exact instantiated patterns. The strongest possible attack would be that the attackers consider ALL possible patterns when generating the adversarial examples. However, this is not possible. Failing all patterns takes extremely long time, and may not even converge. For example, under our randomization setting, the total number of patterns (resizing + padding) is 12528. Thus, instead of choosing such a large number, we choose 21 representative patterns in our ensemble attack scenario, which becomes computationally manageable. Increasing the number of ensembled patterns means: (1) more computation time (take C&W for example, it takes around 0.56 min to generate an adversarial example under vanilla attack, but takes around 8 min to generate an adversarial example under ensemble attack); (2) more memory consumption (at most an ensemble of 30 different patterns can be utilized as one batch to generated adversarial examples for one 12GB GPU, more patterns indicates more GPUs or the GPU with larger memory); (3) larger magnitude of adversarial perturbation.
>
> “biased towards the published adversarial attacks”: Our defense method is not trained using any adversarial examples, so we don’t think it is biased towards any attacks. We extensively test our method on the most popular attacks (one single-step attack FGSM, and two representative iterative attacks DeepFool and C&W), with various network structures, and using large-scale ImageNet datasets. Moreover, we submit this method to a public adversarial defense challenge. Our method is evaluated against 156 different attacks and we are ranked Top 2, which indicates the effectiveness of our method.
>
> “particular problem (e.g. classification) and additive noise”: Currently most works on this topic focus on classification problem and assume additive noise as adversarial perturbation. We follow this setting in this paper. We have two future directions to explore: 1) apply randomization to other vision tasks, 2) apply randomization to other types of attack instead of additive noise. Thanks for the comments.

---

### Official Review · AnonReviewer3 · 2017-11-30
**Simple new baseline (/additional evaluation technique) for defenses against adversarial attacks.**

**Rating:** 7
**Confidence:** 4

**Review:**

This paper proposes an extremely simple methodology to improve the network's performance by adding extra random perturbations (resizing/padding) at evaluation time.

Although the paper is very basic, it creates a good baseline for defending about various types of attacks and got good results in kaggle competition.

The main merit of the paper is to study this simple but efficient baseline method extensively and shows how adversarial attacks can be mitigated by some extent.

Cons of the paper: there is not much novel insight or really exciting new ideas presented.

Pros: It gives a convincing very simple baseline and the evaluation of all subsequent results on defending against adversaries will need to incorporate this simple defense method in addition to any future proposed defenses, since it is very easy to implement and evaluate and seems to improve the defense capabilities of the network to a significant degree. So I assume that this paper will be influential in the future just by the virtue of its easy applicability and effectiveness.

---

> ### Author Response · Authors · 2017-12-30
> **Thanks for pointing out the potential influence of our work**
>
> Thank you very much for the appreciation of our work. The method is indeed simple and effective. Although the randomization idea is not new, we in this paper apply it to mitigate adversarial effects at test time systematically. And we demonstrate the effectiveness on large-scale ImageNet dataset, which is very challenging. Very few defense papers worked on ImageNet before. We hope our method could be served as a simple new baseline for adversarial example defense in the future works.

---

### Official Review · AnonReviewer2 · 2017-12-01
**Simple idea but seems work in well in some cases**

**Rating:** 6
**Confidence:** 3

**Review:**

The paper basically propose keep using the typical data-augmentation transformations done during training also in evaluation time, to prevent adversarial attacks. In the paper they analyze only 2 random resizing and random padding, but I suppose others like random contrast, random relighting, random colorization, ... could be applicable.

Some of the pros of the proposed tricks is that it doesn't require re-training existing models, although as the authors pointed out re-training for adversarial images is necessary to obtain good results.


Typically images have different sizes, however in the Dataset are described as having 299x299x3 size, are all the test images resized before hand? How would this method work with variable size images?

The proposed defense requires increasing the size of the input images, have you analyzed the impact in performance? Also it would be good to know how robust is the method for smaller sizes.

Section 4.6.2 seems to indicate that 1 pixel padding or just resizing 1 pixel is enough to get most of the benefit, please provide an analysis of how results improve as the padding or size increase.

In section 5 for the challenge authors used a lot more evaluations per image, could you provide how much extra computation is needed for that model?

---

> ### Author Response · Authors · 2017-12-30
> **Thanks for your comments and additional experiments are performed**
>
> Thank you very much for the comments, which significantly improve the quality of our paper. We have conducted additional experiments to answer the concerns. These experiments results are included as appendix in the updated paper.
>
> “Other operations”: Yes, other random operations also apply. We tried four operations separately:  random brightness, random contrast, random saturation, and random hue. For each individual operation, we add it to the beginning of the original classification network. We found that these operations nearly have no hurts on the performance of clean images (shown in table 7), but they are not as effective as the proposed randomization layers on defending adversarial examples (shown in table 8-11). By combining these random operations with the proposed randomization layers, the performance on defending adversarial examples can be slightly improved. We have updated these new results in the Appendix A.
>
> “resized beforehand”: Yes, the test images are resized beforehand. There are two reasons: (1) easy to form a batch (e.g., one batch contains 100 images) for classification; (2) stay aligned with the format of the public competition, where the test dataset are all of the size 299x299x3. For the images with variable sizes, we can first resize them to 299x299x3, and then applied the proposed method to defend adversarial examples.
>
> “impact of size in performance”: Adding two randomization layers (increasing size from 299 to 331) slightly downgrades the performance on clean images, as shown in Table 1. This decrease becomes negligible for stronger models. In addition, we also tried applying randomization to smaller-sized images. Specifically, we first resize the images to a size randomly sampled from the range [267, 299), and then randomly pad it to 299x299x3. We evaluate the performance on both the 5000 clean images and the adversarial examples generated under the vanilla attack scenario (shown in table 12). We see that the randomization method works well with smaller sizes, but using larger sizes produces slightly better results. We hypothesize that this is because resizing an image to smaller sizes may lose some information. We have updated the new results in the Appendix B.
>
> “padding or resizing increase”: As the padding size or resizing size increase, there will be a lot more random patterns. So it becomes much harder for the attackers to generate the adversarial example that can fail all the patterns at the same time. Thus, larger size and more paddings will significantly increase the robustness. Notice that the motivation for the experiments in Sec 4.6 is to decouple the effect of padding and resizing. We want to show that (1) adversarial example generated on one padding pattern is hard to transfer to another padding pattern; (2) adversarial example generated on one size is hard to transfer to another size. Using 1-pixel padding and resizing provide a controllable way to verify these two points.
>
> “multiple iterations per image”: The computation time increases linearly with number of iteration per image (e.g., 30x time in our challenge submission). We argue that one iteration is enough to get the most benefits, and additional evaluations only provide marginal gain (as shown in figures 3-5), which is good for the challenge.  The experiments that show the relationship between the classification performance and iteration number is included in Appendix C.

---

### Public Comment · (anonymous) · 2017-11-12
**evaluations are not convincing**

Mitigating adversarial manipulations to deep neural networks via randomization is a promising direction. However, I found evaluations in this paper are not convincing.

1. High-confidence transferable adversarial examples generated by C&W attacks are not evaluated. The paper only evaluated basic C&W attacks.

2. The paper did not compare with recent randomization-based defense. For instance, the paper did not compare with the following paper "Mitigating Evasion Attacks to Deep Neural Networks via Region-based Classification", arxiv 2017.

3. The method proposed in this paper decreases classification accuracy for normal examples in order to increase robustness against adversarial examples. However, there already exists defense that does not decrease classification accuracy for normal examples, but has the same or even better robustness than the proposed method.

---

> ### Author Response · Authors · 2017-11-12
> **Re: evaluations are not convincing**
>
> Thanks for your comments.
>
> (1) C&W attacks is a strong attack, and we follow other papers, e.g., [1], to evaluate basic C&W attacks at current stage. Furthermore, the attacks scenarios considered here are much stronger than black-box attack, while other papers have not studied these before. We will conduct experiments to see how our defense model performs under vanilla attack, single-pattern attack and ensemble attack when confidence increases.
>
> We want to highlight our defense is evaluated on large-scale real image dataset, e.g., ImageNet, which is much harder than defense on small dataset, like MNIST and CIFAR. Meanwhile, the conclusions on small dataset may not be valid on large dataset. For example,  adversarial training helps model get better performance on MNIST, but causes performance to drop on ImageNet (see table 1 at [2])
>
> (2) To the best of my knowledge, there are no randomization-based defense methods available on ImageNet (except some concurrent submissions at ICLR). If you know such reference on ImageNet, please send it to us.
>
> (3) We are not aware of such defense on ImageNet. If you know such reference on ImageNet, please send it to us. Meanwhile, the performance drop on clean images (see table 1) of our best defense model, ens-adv-Inception-ResNet-v2, is only from 100% to 99.2%, which is an acceptable degradation.
>
> [1] Feinman, Reuben, et al. "Detecting Adversarial Samples from Artifacts." arXiv preprint arXiv:1703.00410 (2017).
> [2] Kurakin, Alexey, Ian Goodfellow, and Samy Bengio. "Adversarial machine learning at scale." arXiv preprint arXiv:1611.01236 (2016).

---

> > ### Public Comment · (anonymous) · 2017-11-12
> > **evaluations are not convincing**
> >
> > I still think it is problematic if you do not evaluate high-confidence transferable adversarial examples generated by C&W attacks. Since you use randomization, the model is no longer differentiable. Therefore, high-confidence transferable adversarial examples should be used to attack the defense. If such adversarial examples are not evaluated, the experimental results may be misleading. You can take a look at this paper:
> >
> > Adversarial Examples Are Not Easily Detected: Bypassing Ten Detection Methods.
> >
> > The reference [1] you pointed out for detecting adversarial examples is actually broken.
> >
> > Also, it is useful to compare with state-of-the-art defense instead of adversarial training alone, because adversarial training is known to be not robust for state-of-the-art attacks.

---

> > > ### Author Response · Authors · 2017-11-13
> > > **Re: evaluations are not convincing**
> > >
> > > Thanks for your comments.
> > >
> > > First of all, we would like to highlight two important things in our work. 1). This work is done on large-scale datasets like ImageNet, and only a few defenses (including adversarial training) have demonstrated the effectiveness before. Though MNIST is an interesting dataset on which to test defense ideas, the conclusions may not be readily applied to ImageNet. 2). The attack scenarios considered in our paper are much stronger than black-box attacks. I.e., the network structures and parameters are completely known by the attackers.
> > >
> > > In the paper, we demonstrate the effectiveness of our method on basic C&W attacks, which are very challenging already, and were not well studied on ImageNet before. In order to overcome the problem that randomization models are not differentiable, we considered single-pattern attacks and ensemble-pattern attacks in the experiments. The experimental results indicate that adversarial examples generated under ensemble-pattern attacks are stronger than others. Note that the C&W attacks are very slow. Take the basic C&W attack against inception-resnet-v2 for example. It takes ~17 mins to generate adversarial examples for a batch of 30 images under vanilla attack scenario, and takes ~8 mins to generate adversarial examples for 1 image under ensemble-pattern attack scenario. Generating higher-confidence adversarial examples will significantly increase the time consumption even further, and thus, may not be practical. So we focus on basic C&W attacks in our experiments at current stage.
> > >
> > > For the baseline included in this paper, we want to point out three things. 1). To the best of our knowledge, adversarial training is the most effectiveness method on large-scale dataset like ImageNet. We are confused by the words “the state-of-the-art defense”, please refer to it explicitly. 2). The adversarially trained model is not robust to iterative attacks, and is used more like a network backbone rather than baseline in our paper. We combine the adversarially trained model, which is robust to single-step attacks, and randomization layers, which improve network robustness to iterative attacks, together to form our best defense model. 3). There are 100+ defense teams and 150+ attacks teams participate in this public adversarial defense challenge, and our model is ranked top 2. We argue that this challenge provides us sufficient baselines (including very strong ones) to compare with, which convincingly demonstrates the effectiveness of our method in real world scenario.

---

### Author Response · Authors · 2017-11-12
**update on public challenge evaluation results**

Our submission ranked Top 2 (among 100+ teams) at the final round of a public adversarial defense challenge, where the number of test images is increased to 5000, and the number of different attack methods is increase to 150+. It reached a normalized score of 0.924, which is far better than using adversarial training alone with a normalized score of 0.773 (ranked No.56).

We will reveal the URL of the challenge once the revision period is over.

---

### Author Response · Authors · 2017-12-30
**General reply to all reviewers**

We would like to thank the reviewers for their thoughtful responses, and are glad to see that there is a consensus among the reviewers to accept this work. In order to address the concerns from reviewers, we have conducted more experiments (shown in Appendix) and updated the paper to describe the experiments more clearly. We are grateful to each of the reviewers to help us improve the work. Please find individual replies to each of the reviews in the respective threads.

---

### Decision · Program_Chairs · 2018-01-29
**ICLR 2018 Conference Acceptance Decision**

**Decision:**

Accept (Poster)

**Comment:**

Paper proposes adding randomization steps during inference time to CNNs in order to defend against adversarial attacks.

Pros:

- Results demonstrate good performance, and the team achieve a high rank (2nd place) on a public benchmark.
- The benefit of the proposed approach is that it does not require any additional training or retraining.

Cons:

- The approach is very simple, common sense would tend to suggest that adding noise to images would make adversarial attempts more difficult. Though perhaps simplicity is a good thing.
- Update: Paper does not cite related and relevant work, which takes a similar approach of requiring no retraining, but rather changing the inference stage: https://arxiv.org/pdf/1709.05583.pdf

Grammatical Suggestions:

This paper would benefit from polishing. For example:

- Abstract: sentence 1: replace “their powerful ability” to “high accuracy”
- Abstract: sentence 3: replace “I.e., clean images…” with “For example, imperceptible perturbations added to clean images can cause convolutional neural networks to fail”
- Abstract: sentence 4: replace “utilize randomization” to “implement randomization at inference time” or something similar to make more clear that this procedure is not done during training.
- Abstract: sentence 7: replace “also enjoys” with “provides”

Main Text: Capitalize references to figures (i.e. “figure 1” to “Figure 1”).

Introduction: Paragraph 4: Again, please replace “randomization” with “randomization at inference time” or something similar to better address reviewer concerns.